# Energy Balance Closure in the Tugai Forest in Ebinur Lake Basin, Northwest China

Dexiong Teng [1,2] , Xuemin He [1,2], Lu Qin [3] and Guanghui Lv [1,2,*]

1   College of Resources and Environment Science, Xinjiang University, Urumqi 830046, China; shadows923@163.com (D.T.); hxm12345516@163.com (X.H.)
2   Key Laboratory of Oasis Ecology of Ministry of Education, Xinjiang University, Urumqi 830046, China
3   College of Tourism, Xinjiang University, Urumqi 830046, China; ql198588@163.com
*   Correspondence: ler@xju.edu.cn

**Abstract:** A persistent problem in surface flux research is that turbulent fluxes observed by eddy covariance methods tend to be lower than the available energy. Using 7 years of eddy covariance flux observations in the Ebinur Lake National Wetland Nature Reserve (ELNWNR) in Xinjiang, Northwest China, this study analyzes the surface–atmosphere energy transfer characteristics at the station to explore variation characteristics of the energy flux and the energy balance closure (EBC), and the factors that influence EBC. The results show that: (1) diurnal and seasonal variations are observed in turbulent flux, available energy, and the partitioning of sensible and latent fluxes affected by environmental factors; (2) the degree of EBC varies significantly diurnally and seasonally, with EBC during the growing season significantly higher than during the dormant season; (3) due to the surface heterogeneity, EBC exhibits significant variations with wind direction that differ between the growing and dormant seasons; (4) environmental factors (e.g., vapor pressure deficit and air temperature) are important in limiting near-surface EBC, but they play a secondary role compared with the state of atmospheric motion. This study provides a basis for accurately assessing the material and energy exchanges between the desert Tugai forest ecosystem and the atmosphere.

**Keywords:** energy balance closure; eddy covariance method; Tugai forest; turbulence flux

## 1. Introduction

The observation and research of near-surface turbulent fluxes (sensible and latent heat fluxes) and available energy (net radiation and soil heat fluxes) occupy an extremely important position in atmospheric boundary layer meteorology [1]. Evaluating the energy balance closure of terrestrial ecosystems is an important part of research on ecosystem–atmosphere interactions, because the energy balance affects the regional climate and water balance. Understanding the energy balance is also important for evaluating ecosystem function. In the context of global climate change, the structures and functions of terrestrial ecosystems are constantly changing. More complete knowledge of the energy balance will deepen our understanding of ecosystem variations and provide more information on how ecosystems respond to global changes.

Flux observation data from eddy covariance (EC) measurements have been widely used to validate remote sensing flux products, to develop and validate land-surface process models, and to inform global energy and material cycle research [2,3]. However, many observational studies have found that the sum of the sensible heat flux (H) and latent heat flux (LE) observed at most EC sites around the world is lower than the available energy (the difference between the net radiation Rn and the surface heat storage) [4–7]. The difference can reach 10%–30% of the available energy [8–10]. This underestimation is called the "EC energy balance closure (EBC) problem". Numerous studies have generated important insights into the problem of EC energy balance closure but they have not provided conclusive answers about the cause and solution of the problem.

To study the causes of near-surface energy observation imbalance, a large-scale Energy Balance Experiment (EBEX) test was organized in 2000 [5,6,11]. The conclusion of this experiment was that it is difficult to reach energy balance using near-surface observations. Thus, the emergence of the EC energy balance closure problem calls into question the accuracy and reliability of EC measurements and increases the uncertainty of research that is based on EC flux observations. At present, the problem of EC energy balance closure has become a bottleneck in micrometeorology and the study of surface–atmosphere interaction. This problem also hinders the improvement of energy, water, and carbon exchange algorithms; slows the improvement of land surface process models; and reduces the reliability of climate forecasts.

Substantial research has been carried out on the problem of surface energy balance closure [11–15]. Nevertheless, the imbalance of surface energy remains a difficult problem in the study of surface–atmosphere exchange. High- and low-frequency loss of turbulence will cause EC to underestimate the turbulent flux [16,17]. However, energy balance non-closure does not necessarily constitute evidence for erroneous turbulent flux measurements [17]. In areas with substantial terrain heterogeneity, local circulations can form and nocturnal disturbances may occur, making the energy balance difficult to close [18]. Horizontal and vertical advection also substantially impact the surface energy balance [5,19,20]. Although many methods have been proposed to correct the surface energy imbalance, these methods have not been fully successful [21–23]. The study of surface energy balance using multi-tower observations shows that the method of spatial averaging cannot solve the problem of energy balance non-closure [23]. Furthermore, capturing the impact of surface heterogeneities on surface–atmosphere energy exchanges remains challenging. Heterogeneity-induced secondary circulations can contribute substantially to vertical heat and moisture transport beyond the turbulent processes captured by EC methods. Thus, they may contribute to incomplete energy balance closure [4]. Some preceding studies demonstrated that EBC was a function of environmental factors, which has been confirmed by many observational studies [24–26]. One key to solving the energy balance closure problem is to find the factors and the physical processes affecting the near-surface energy balance; another is to quantitatively analyze these factors and processes.

As the largest terrestrial biome on the planet, forests have a significant impact on the redistribution of energy in the regional and global climate. EC technologies based on micrometeorology provide opportunities to study the energy balance in forest ecosystems. In recent years, numerous studies have used flux towers to analyze energy flux characteristics and energy balance closure in tropical, subtropical, boreal, and temperate forests [10,27–29]. However, there are few studies on energy fluxes in riparian forests, especially riparian forests in arid desert regions. The arid desert riparian forest is of great importance to regional water and energy exchanges [30]. The Tugai forest is a typical type of desert forest. Its special thermal cycling characteristics—oasis-desert effect and dewfall—profoundly affect the ecological environment where it is located [31,32]. The aerodynamic roughness length is investigated to quantify the complexity of each wind direction, in order to evaluate the influence of the heterogeneous underlying surfaces on energy balance closure. More complete knowledge of the energy balance characteristics of the Tugai forest will enhance the understanding of the interaction between surface processes and climate change, and provide a reference for studying the vulnerability and adaptability of desert ecosystems to climate change.

In this study, we use EC observation data obtained in Ebinur Lake National Wetland Nature Reserve (ELNWNR), Xinjiang, China, from January 2012 to April 2019, combined with synchronous micrometeorological data. Our objectives were to compare the influence of micrometeorological parameters and environmental factors on energy balance closure. We: (1) analyzed the diurnal and seasonal variations of the energy flux components over a Tugai forest, (2) evaluated the seasonal difference in energy balance closure at a site with physical heterogeneity, and (3) analyzed the relationships between the EBC values and the related meteorological and environmental variables.

## 2. Materials and Methods

### 2.1. Study Site Description

The study area is located in ELNWNR, Xinjiang, China (44°37′05″–45°10′35″ N, 82°30′47″–83°50′21″ E), which has a northern, temperate, continental, dry climate and an altitude of 190 m. The average annual temperature in the Ebinur Lake basin is 6−8 °C, annual precipitation is about 100 mm, annual evaporation is about 1600 mm, the groundwater depth is 1.8−2.7 m, the annual effective radiation is about $5.694 \times 10^9$ W m$^{-2}$ year, and the annual sunshine duration is about 2800 h. The frost-free period is approximately 160 d and the annual average relative humidity is 50%. The salt content of the soil surface layer during the growing season is about 8.80 g kg$^{-1}$. The flux tower is located in Tugai forest (Figure 1d), approximately 100 m from the Aqikesu River (44°37′4.8″ N, 83°33′59.4″ E). The dominant plant species at this location are *Populus euphratica* Oliv. (*Populus euphratica*), *Haloxylon ammodendron* Bunge (*Haloxylon ammodendron*), and common reed (*Phragmites australis*), with a combined coverage rate of more than 60% during the growing season [33]. These species are accompanied by salt spike plants, salt knot plants, *Suaeda*, and other desert-specific short-lived plants. The mean canopy height of *Populus euphratica* is approximately 8.5 m.

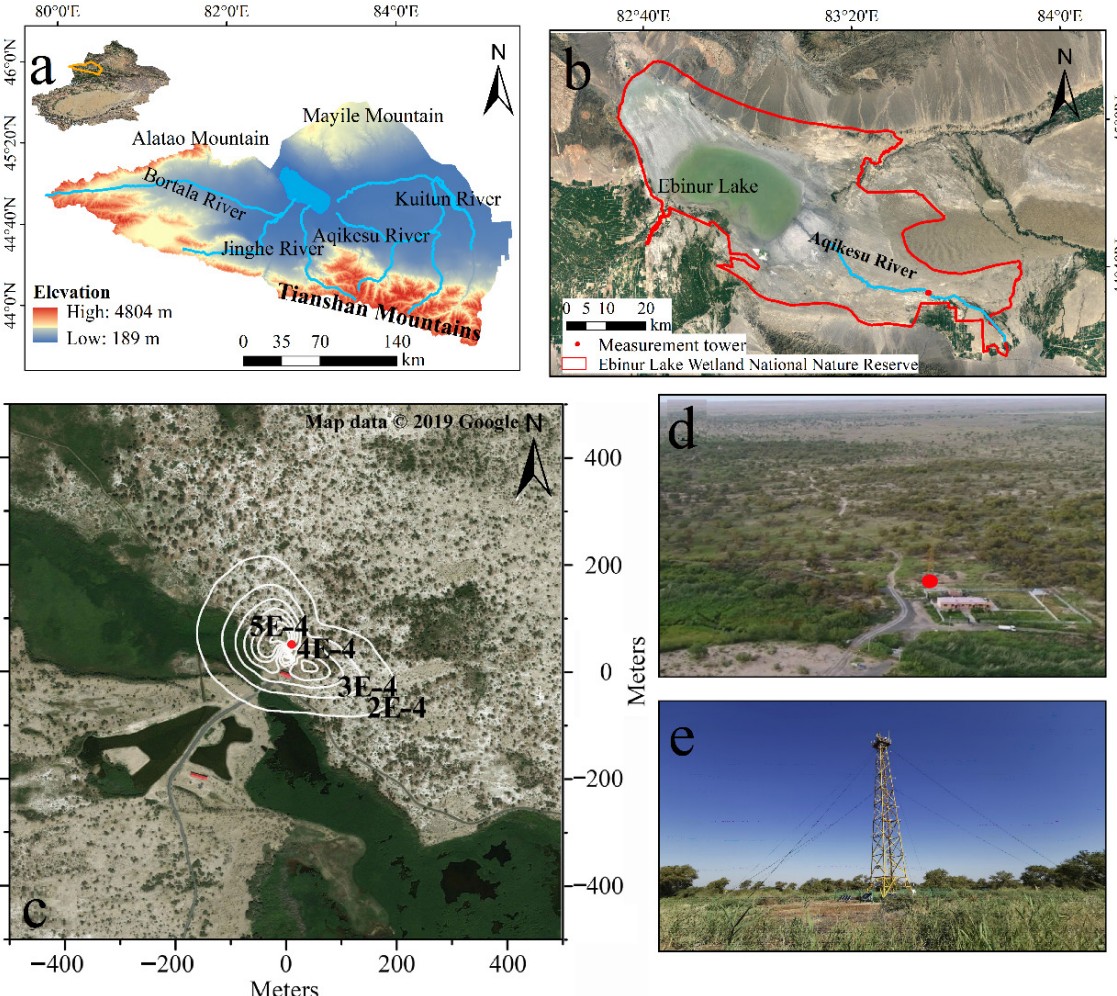

**Figure 1.** The measurement site in this study. (**a**) Location map of the Ebinur Lake basin; (**b**) location of the measurement tower within the region; (**c**) remote sensing image in June, with the average footprint for the entire observation period overlaid [34]; (**d**) aerial photo of the tower site in May; and (**e**) photograph of the tower site in August [35]. The measurement tower is marked with a red circle. White lines represent isopleths of average flux contributions to the measured values.

## 2.2. Measurements and Data

Sensible heat flux (H) and latent heat flux (LE) were continuously measured by an open-path EC system from January 2012 to April 2019. The EC observation system was located at 15 m above the ground and the outrigger faced 225°. The system included a 3D ultrasonic anemometer (CSAT3, Campbell Scientific Ltd., Logan, UT, USA) and an infrared $CO_2/H_2O$ analyzer (EC150, Campbell Scientific Ltd., Logan, UT, USA). Soil heat flux (G; W m$^{-2}$) was measured by a soil heat flux plate (HFP01, Campbell Scientific Ltd., Logan, UT, USA) buried 5 cm underground. The 4 components of net radiation (Rn; W m$^{-2}$), upward and downward short-wave radiation and long-wave radiation, were independently observed by a 4-component net radiometer (NR01, Campbell Scientific Ltd., Logan, UT, USA) with a mounting height of 9 m. The weather observation system comprised an air temperature and humidity sensor (HMP155A−L, Campbell Scientific Ltd., Logan, UT, USA), wind speed sensor (010C−1, Met One Instruments Inc., Grants Pass, OR, USA), wind direction sensor (020C−1, Met One Instruments Inc., Grants Pass, OR, USA), barometric pressure measuring instrument (CS100, Campbell Scientific Ltd., Logan, UT, USA), and a precipitation sensor (TE525MM, Campbell Scientific Ltd., Logan, UT, USA). These instruments automatically recorded regular meteorological data (e.g., average wind speed, temperature, air pressure, and net radiation) every 30 min.

## 2.3. Data Processing and Post Processing

In this study, EddyPro6.2.1 software (LI-COR Inc., Lincoln, NE, USA) was used to calculate 30-min averages of turbulent fluxes. In the calculation script, calculation steps, such as outlier removal, tilt correction (using the double rotation for coordinate rotation); time lag correction, sonic virtual temperature correction, and Webb, Pearman, and Leuning (WPL) correction [36], were performed. Next, the 30-min flux data were evaluated for data quality using tests, such as the turbulence stability test and the development test; data with poor evaluation results were excluded. Flux data observed during times with a wind direction of 30° or 60° were also excluded to avoid effects from the flux tower structure. Because precipitation can affect the sensible heat flux (H; W m$^{-2}$) and latent heat flux (LE; W m$^{-2}$), data measured during precipitation were eliminated. The daily growing degree-days (GDD) were used to determine the start and end times of the growing season and dormant season. The formula for calculation is GDD = $T_{average} - T_{base}$, where $T_{average}$ is the daily average air temperature and $T_{base}$ is the base temperature (6 °C in this study). Data processing and statistical analysis were performed in R language.

## 2.4. Energy Balance Closure

In energy balance studies, the surface energy balance equation in W m$^{-2}$ can be expressed as follows:

$$H + LE = Rn - G - S - Q \tag{1}$$

where H is the sensible heat flux; LE is the latent heat flux, Rn is the net radiation, G is the shallow soil heat flux, S is the canopy heat storage, and Q is the additional energy, (the sum of additional sources and sinks). The additional energy source/sink term Q may include vertical flux divergence, horizontal advection, photosynthesis of plants, and the process of water absorption from plants to their leaves. These quantities are difficult to determine due to the constraints of observation; however, they are usually relatively small. Thus, the additional energy source/sink term Q was ignored in this study. This study uses standard major axis regression (SMA) to comprehensively analyze the energy balance closure at this site. The slope coefficient from the SMA is used to evaluate EBC. In an ideal energy balance, the SMA return line of available energy (Rn − G − S) and turbulent flux (H + LE) should both have a slope of 1 and pass through the origin.

To analyze the factors that affect EBC, variables are computed that include friction velocity ($u_*$), the thermally induced turbulent parameter (TT), atmospheric stability ($z/L$), turbulent kinetic energy (TKE), relative vertical turbulent intensity (RI), and the correlation

coefficients of vertical wind velocity with water vapor density ($R_{wq}$) and temperature ($R_{wT}$) over 30 min. The definitions of $u_*$, TT, $z/L$, TKE, RI, $R_{wT}$ and $R_{wq}$ are as follows:

$$u_* = \left(\overline{u'w'}^2 + \overline{v'w'}^2\right)^{0.25} \tag{2}$$

$$TT = \kappa \frac{g}{\theta} \overline{\theta' w'} \tag{3}$$

$$\frac{z}{L} = -z \frac{TT}{u_*^3} \tag{4}$$

$$TKE = 0.5\left(\overline{u'}^2 + \overline{v'}^2 + \overline{w'}^2\right) \tag{5}$$

$$RI = \frac{\sqrt{\overline{w'}^2}}{\sqrt{\overline{w'}^2} + U'} \tag{6}$$

$$R_{wq} = \frac{\overline{w'q'}}{\sigma_w \sigma_q}, \tag{7}$$

$$R_{wT} = \frac{\overline{w'T'}}{\sigma_w \sigma_T}, \tag{8}$$

where $u'$, $v'$, and $w'$ are three-dimensional wind speed components; $\sigma_u$, $\sigma_v$, and $\sigma_w$ are the corresponding standard deviations of the three-dimensional wind speed components; $\kappa$ is the von Karman constant; $g$ is the acceleration of gravity; $\theta$ is the virtual temperature; $T$ is the air temperature; $q$ is the water vapor density; and $U$ is the average horizontal wind speed. Additionally, we examined the relationship between specific environmental variables and EBC. Vapor pressure deficit (VPD) is closely related to evapotranspiration from vegetation.

The EC system has a time lag in flux observations to a certain extent, while the meteorological and environmental observations are relatively real-time. Therefore, it is necessary to examine the impact of environmental variation on energy balance closure. Variations in air temperature ($T_{air}$) and relative humidity (RH) are closely related to the turbulent flux and available flux, respectively. The difference in $T_{air}$ between each 30-min averaging period and the previous period ($\Delta T_{air}$) represents $T_{air}$ variation with time. Similarly, the difference in RH between each 30-min averaging period and the previous period ($\Delta$RH) represents RH variation with time.

### 2.5. Aerodynamic Roughness Length

The aerodynamic roughness can be used to characterize the aerodynamic properties of vegetation-covered surfaces, and it is the basic parameter to be determined to study the process of energy and material exchange between vegetation and the atmosphere. In the neutral atmospheric condition, the wind velocity profile on the vegetation-covered surface is expected to be logarithmic and takes the following form:

$$\overline{u} = \frac{u_*}{\kappa} \log\left(\frac{z-d}{z_0}\right) \tag{9}$$

where $\overline{u}$ is the wind speed at height $z$, $d$ is the displacement height $d$, $z_0$ is the aerodynamic roughness length, and $\kappa$ is the von Karman constant.

### 2.6. Footprint Calculation

Footprint provides a quantitative tool to describe the source area or effective fetch of surface–atmosphere exchange measurements. In this study, a flux footprint model [34] was used to analyze the source area of turbulent fluxes measured at the tower site. This model can be applied to data analyses of long-term measurements. It is mainly used to calculate

the crosswind integral function of the flux footprint. The required parameters include the observation height, canopy height, friction velocity, and wind direction.

## 3. Results

### 3.1. Meteorological Conditions and Footprint

Over the study period (2012–2019), the interannual dynamics of daily mean air temperature (Ta), vapor pressure deficit (VPD), and global radiation (Rg) showed a clear seasonal cycle between the growing and dormant seasons, with higher values during the growing season (Figure 2a–c). Daily relative humidity (RH) was higher during the dormant season than during the growing season (Figure 2d). During the study period, mean Ta was 8.83 °C, ranging from −28.04 °C to 34.13 °C. The VPD ranged from 0.02 to 4.17 kPa during the same period; the maximum and minimum VPD values occurred in July and January, respectively. It is important to note that more than half of the precipitation data are missing due to instrument failure. Precipitation in the Tugai forest is sparse and uneven, while annual evaporation greatly exceeds annual precipitation. Although vegetation growth in most arid desert regions is primarily affected by precipitation, the Tugai forest is not, as its main water sources are rivers.

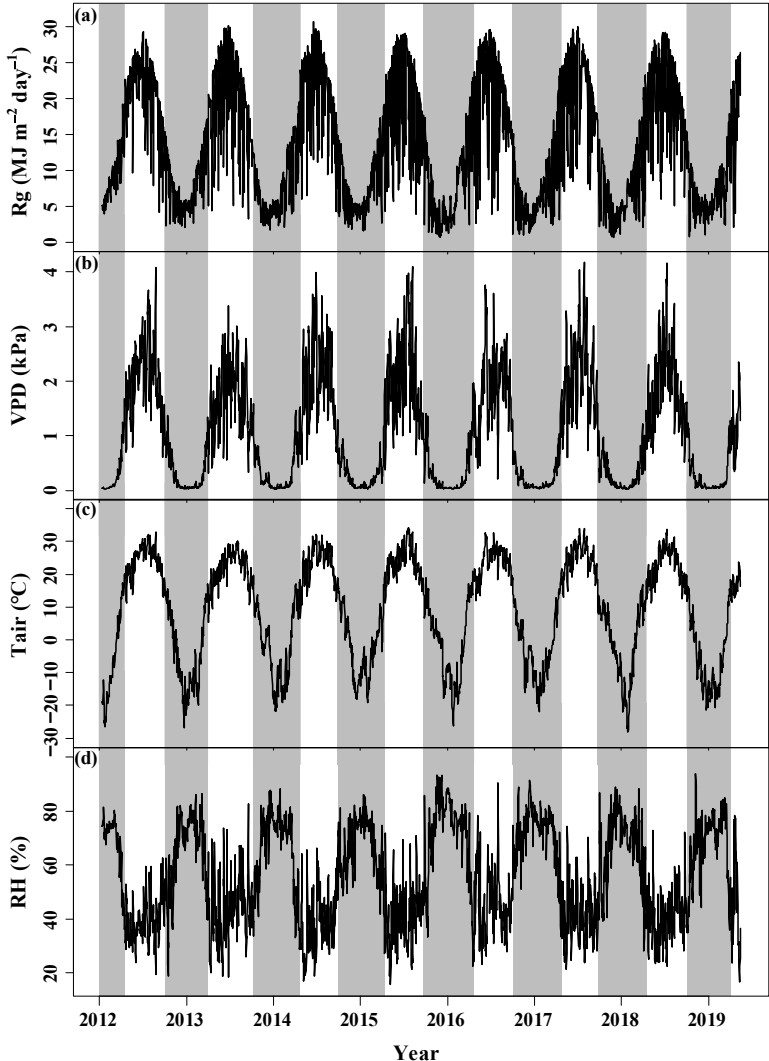

**Figure 2.** Meteorological conditions during the study period: (**a**) global radiation (Rg), (**b**) vapor pressure deficit (VPD), (**c**) air temperature (Tair), and (**d**) relative humidity (RH). White and gray shading indicates the growing season and the dormant season, respectively.

Analysis of wind speed and wind direction forms the basis of research on the contributing source area of fluxes, as wind greatly influences the areas that contribute to measured fluxes. As depicted in Figure 3, the contributing source area of EC in the dormant season was larger than that in the growing season, and the shape changed with the prevailing wind directions during each season. The source areas of the EC measurements at the site extended along the prevailing wind direction (Figure 4). During the growing season, the dominant wind directions were east and northwest (Figure 4a). The dominant wind directions during the dormant season were northwest and east.

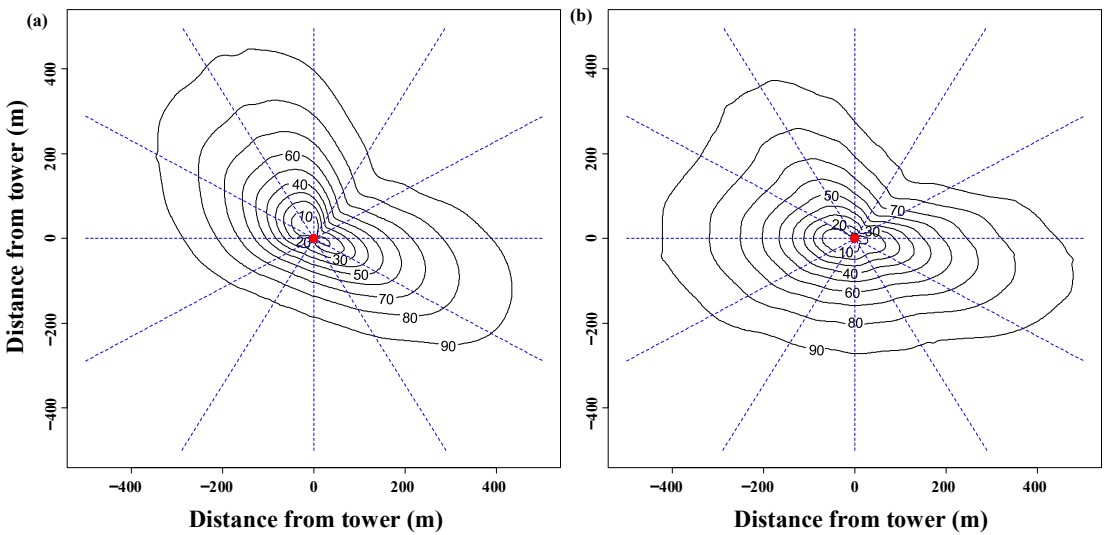

**Figure 3.** Flux footprint during the (**a**) growing season and (**b**) dormant season. The red circle represents the measurement tower; the black solid lines represent isopleths of accumulated percentage flux contributions to measured values.

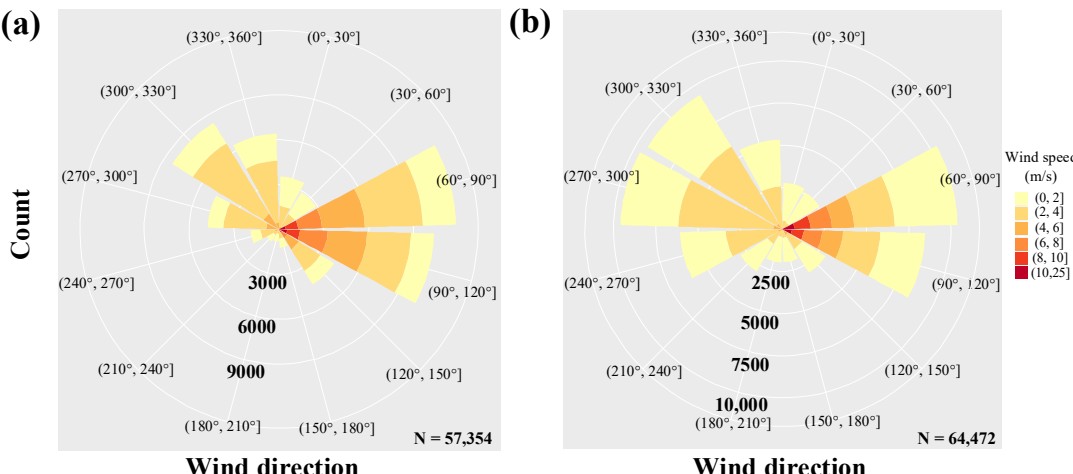

**Figure 4.** Wind direction and cumulative wind speed frequency during the (**a**) growing season and (**b**) dormant season. Colors represent wind speed bins.

### 3.2. Energy Partitioning and Underlying Surface Roughness

Daily cycles of friction velocity (Figure 5) exhibited a unimodal pattern during both the growing and dormant seasons, with the highest values during daylight hours. Throughout the daily cycle, the growing season had higher mean friction velocity values than the dormant season.

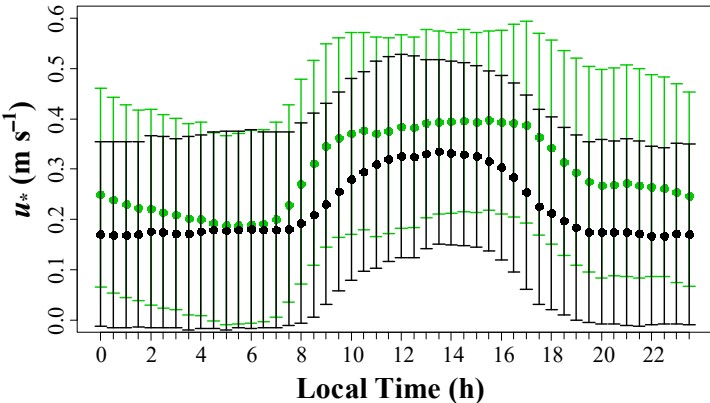

**Figure 5.** Diurnal variation of the mean and standard deviation of friction velocity during the growing season (green) and dormant season (black). Error bars represent the standard deviation of friction velocity.

The average aerodynamic roughness lengths $z_0$ with respect to wind direction were compared in order to examine the underlying surface condition. There was a significant difference in aerodynamic roughness length $z_0$ at the $p = 0.05$ level between different wind direction bins during both the growing and dormant seasons (Figure 6), indicating that the underlying surface is heterogeneous. The most interesting aspect of this graph is that the variation of the aerodynamic roughness length $z_0$ with respect to wind direction bins during the growing seasons was greater than that during the dormant seasons. During both the growing and dormant seasons, the aerodynamic roughness length $z_0$ with respect to wind direction (60°, 120°) from the desert was the smallest ( Figure 1; Figure 6). Thus, when the wind is from those directions, the near-surface airflow is less affected by nearby terrain.

To describe how the components of energy balance are partitioned during the different seasons, the diurnal patterns of the 30-min averages of Rn, LE, H, G, and S in the growing and dormant seasons are plotted in Figure 7. In general, H and LE followed the daytime behavior of solar radiation. During the growing season, daytime LE exceeded H (Figure 7a), indicating that the majority of net radiation was consumed by evapotranspiration. During the dormant season, however, H exceeds LE during daylight hours (Figure 7b). The maximum Rn occurred at around 12:00 h local time during the growing season, while the maximum occurred at around 11:00 h local time during the dormant season.

During the growing season (Figure 8a), EBC values increased between 06:30 h and 16:00 h, before rapidly declining to a minimum value at 6:30. During the dormant season (Figure 8b), EBC values increased between 08:30 h and 17:00 h, and then declined to a minimum value at 08:30 h. EBC values in both seasons increased during daylight hours; thus, values were higher during the afternoon than the morning. This daytime pattern has been observed at many sites during the growing and dormant seasons [37–40]. During both seasons, EBC values at this site were less than 1, indicating that the turbulent flux may have been underestimated.

Figure 9 shows the relationship between Rn − G − S and H + LE during the growing season (Figure 9a) and the dormant season (Figure 9b). The determination coefficient was nearly identical during the two seasons: 0.793 and 0.795 during the growing and dormant seasons, respectively. The intercept value was positive for both seasons: 15.392 W m$^{-2}$ during the growing season and 3.578 W m$^{-2}$ during the dormant season. This difference could be associated with differences in the magnitude of mean turbulent fluxes during different seasons. The slope of the relationship between Rn − G − S and H + LE during the growing and dormant seasons was 0.670 and 0.643, respectively. The significantly smaller slope during the dormant season indicates that EBC values during the dormant season were lower, on average, than those during the growing season. Thus, turbulent flux observations were smaller relative to measured available energy during the dormant season, compared with during the growing season.

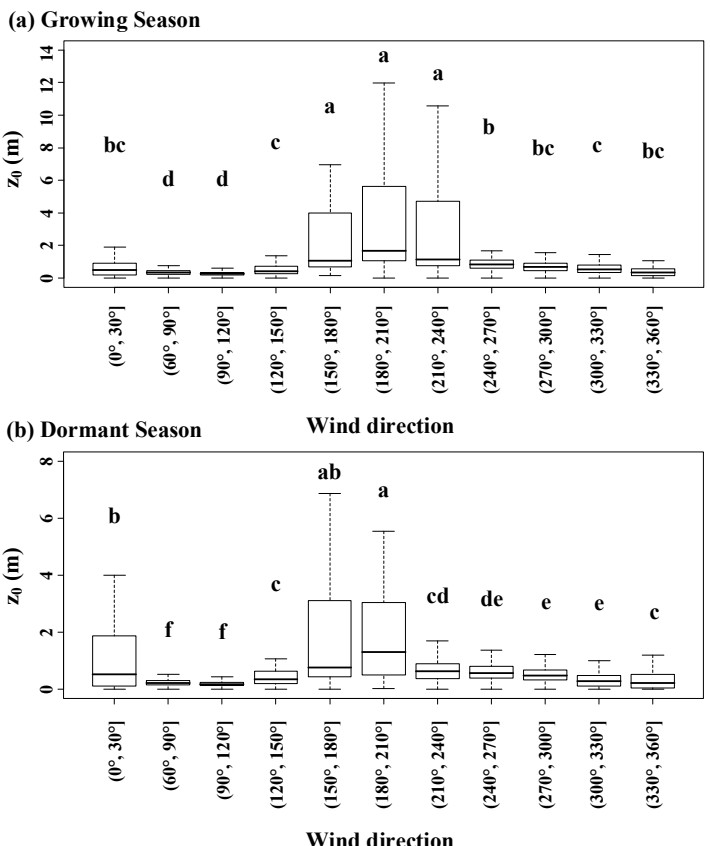

**Figure 6.** Aerodynamic roughness length parameter ($z_0$) for wind direction categories during the (**a**) growing season and (**b**) dormant season. Statistical significance was analyzed using analysis of variance and least significant difference tests as appropriate. Columns with different capital letters are significantly different ($p < 0.05$) according to Fisher's least significant difference test.

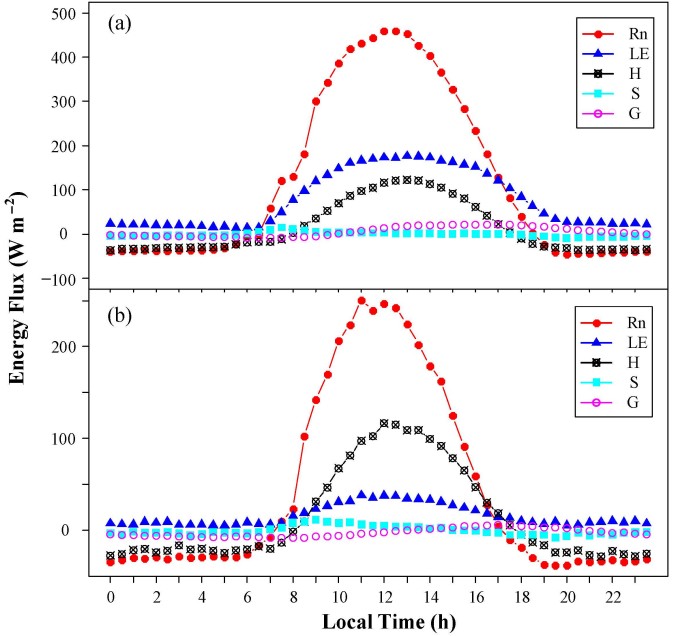

**Figure 7.** Mean daily variation in energy fluxes during the (**a**) growing season and (**b**) dormant season.

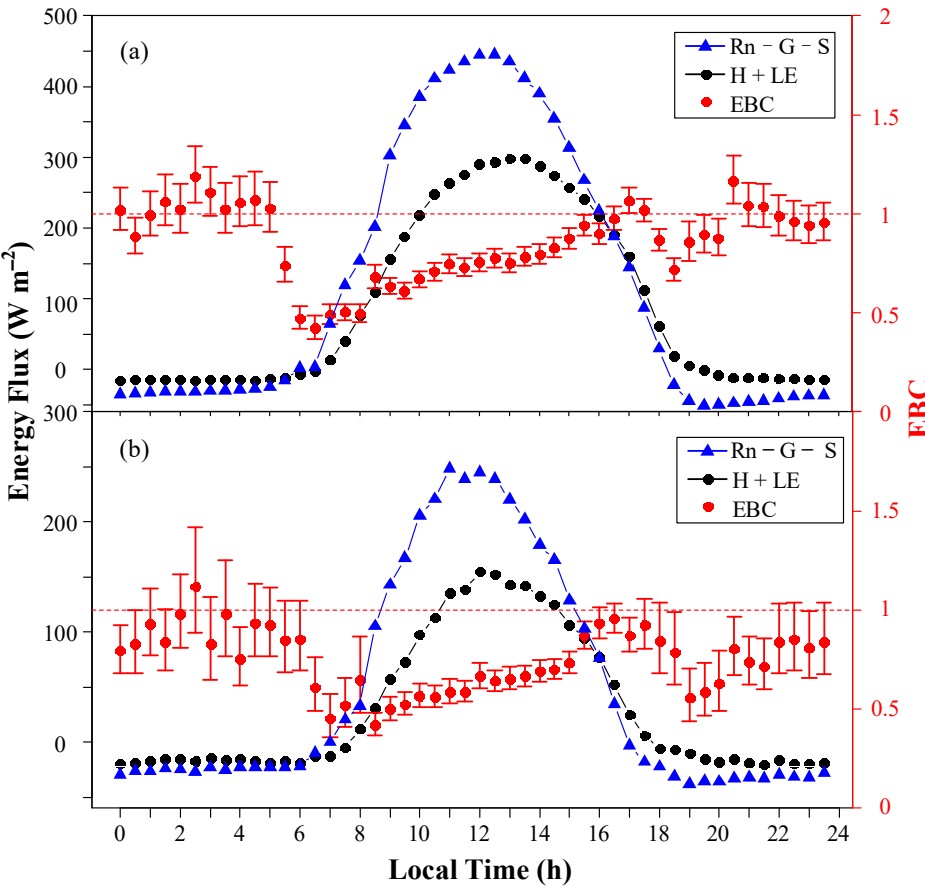

**Figure 8.** Diurnal variations of (left axis) available energy (net radiation (Rn) − shallow soil heat flux (G) − canopy heat storage (S)) and the sum of turbulent fluxes (sensible heat flux (H) + latent heat flux (LE)) for the 30-min averaged measurements, along with (right axis) energy balance closure (EBC) values during the (**a**) growing season and (**b**) dormant season. Red dots indicate the slope of the relationship between available energy and turbulent fluxes, and error bars represent the 95% confidence interval of the slopes from standard major axis regression (SMA).

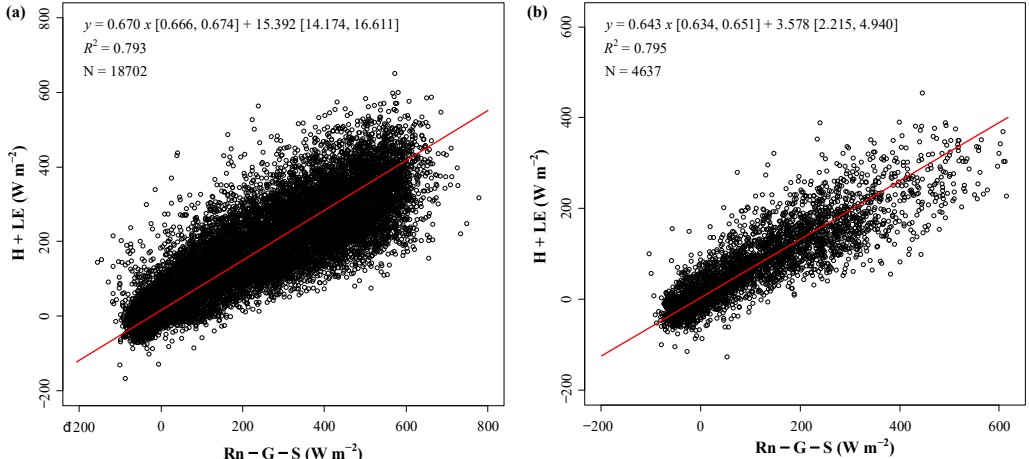

**Figure 9.** Relationship between the available energy in the system (Rn − G − S) and the sum of the sensible and latent heat fluxes (H + LE) during the (**a**) growing season and (**b**) dormant season. Rn is the net radiation; G is the soil heat flux; and S is the canopy heat storage. Solid red lines show the linear fit, and points represent 30-min averages. N is the number of samples included in each regression.

### 3.3. Energy Balance Closure

The EBC values for each category of $R_{wT}$ and $R_{wq}$ were estimated by the slope parameter of the relationship between Rn − G − S and H + LE. Figure 10 shows how EBC varied with $R_{wT}$ and $R_{wq}$ during the growing and dormant seasons. For $R_{wq}$ > 0.12, EBC increased with increasing $R_{wq}$ during the growing season, while EBC values during the dormant season leveled off at $R_{wq}$ > 0.2. However, for $R_{wq}$ < 0.12, there was no straightforward relationship between $R_{wq}$ and EBC values during either season. For $R_{wT}$, EBC during the dormant season reached a minimum value at approximately $R_{wT}$ = −0.05 (Figure 10b). For all $R_{wT}$ values > −0.2, EBC values were significantly higher during the growing season than the dormant season. Variations in $R_{wT}$ and $R_{wq}$ are associated with different EBC values during different seasons.

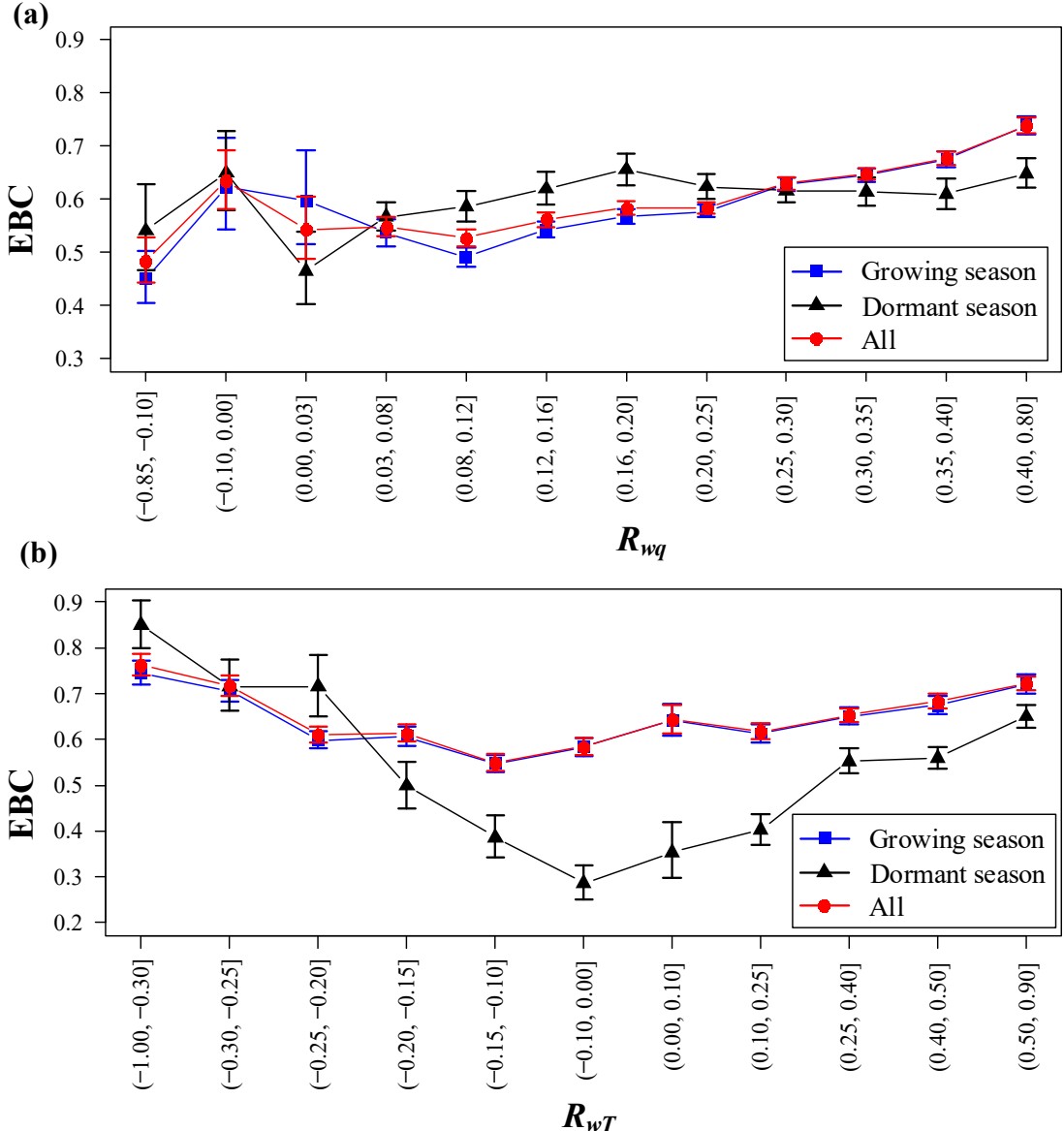

**Figure 10.** Average EBC for each bin of (**a**) water vapor density ($R_{wq}$) and (**b**) water vapor density temperature ($R_{wT}$). Dots indicate the slope of the relationship between available energy (Rn − G − S) and total heat flux (H + LE), and error bars represent the 95% confidence interval of the slopes from SMA.

Atmospheric motion greatly affects surface energy balance closure. Under unstable or strongly turbulent conditions, the surface energy balance closure usually performs comparatively well, while it usually performs comparatively poorly when the atmosphere

is steady and turbulence is weak, such as at night. To test whether the energy balance characteristics of the underlying surface followed this pattern, the relationships of the degree of surface energy closure with atmospheric stability, friction velocity, and relative turbulence intensity were analyzed.

During both seasons, EBC increased with increasing $u_*$ for $u_* < 0.50$ m s$^{-1}$. For $u_* > 0.50$ m s$^{-1}$, EBC values during the growing season remained at approximately 0.7, while EBC values during the dormant season decreased with increasing $u_*$ until $u_*$ was approximately 0.7.

In addition to the friction velocity, thermal factors also affect EBC by affecting the atmospheric stability of the surface layer. During the growing season, except when thermal turbulence is strong (TT > $1.5\times10^{-3}$ or TT < $-5.5\times10^{-4}$), average EBC values tended to be relatively constant (no significant difference between bins, and small intra-bin variation) at approximately 0.6 (Figure 11b). For TT > $-2.0\times10^{-4}$, EBC values were significantly lower during the dormant season than during the growing season. By comparison, the relationship between TT and EBC shows greater seasonal difference than the relationship between $u_*$ and EBC.

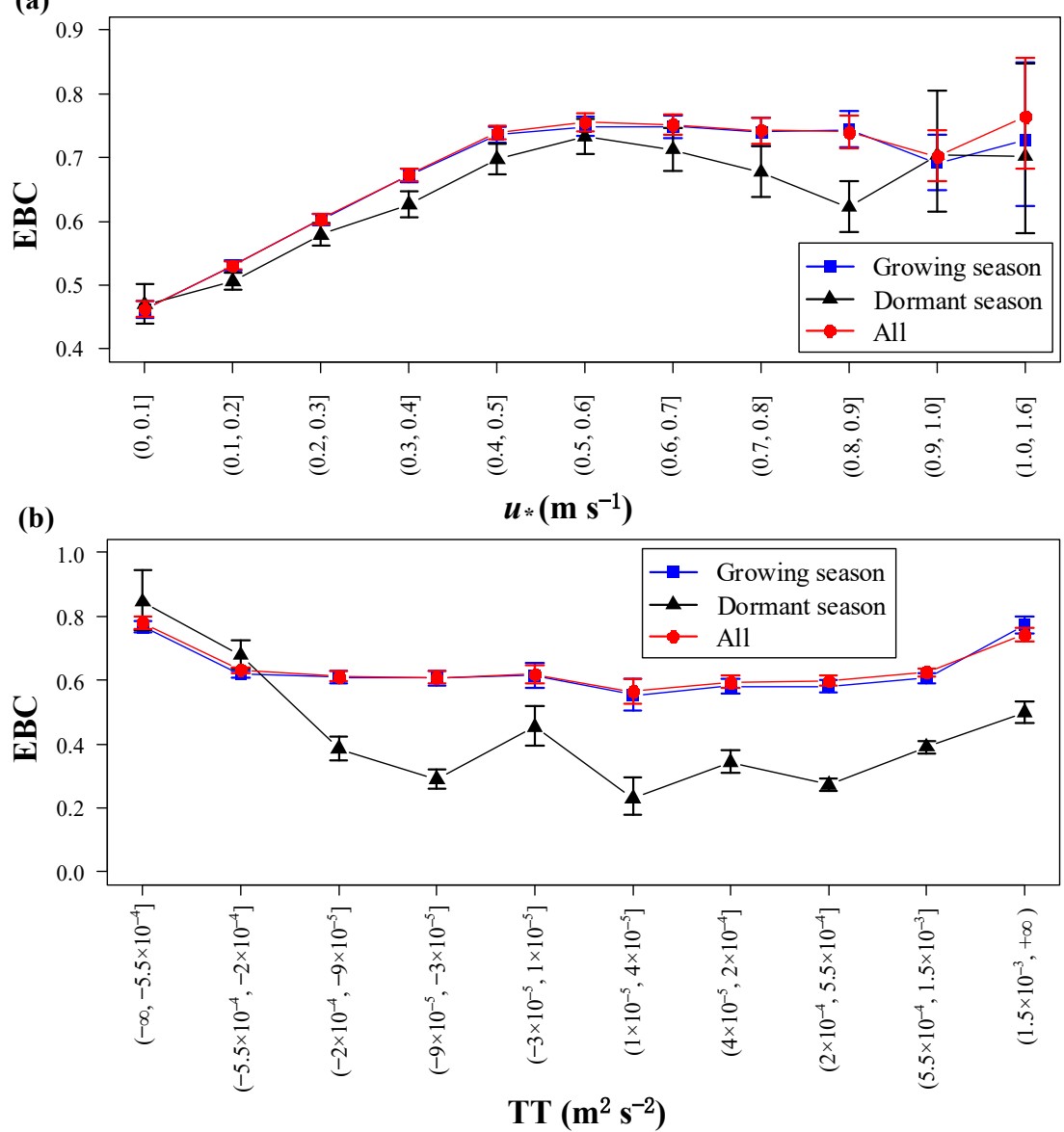

**Figure 11.** Average EBC for each bin of (**a**) friction velocity ($u_*$) and (**b**) the thermally induced turbulent parameter (TT).

The relationship between EBC and atmospheric stability was assessed by using percentiles to group half-hourly LE + H and Rn − G − S data into 11 stability parameter ($z/L$) categories (Figure 12a). When the atmosphere was unstable ($z/L < −0.10$), EBC values increased with increasing $z/L$; when the atmosphere was neutral ($−0.10 < z/L < 0$) or stable ($z/L > 0$), EBC values increased as $z/L$ decreased.

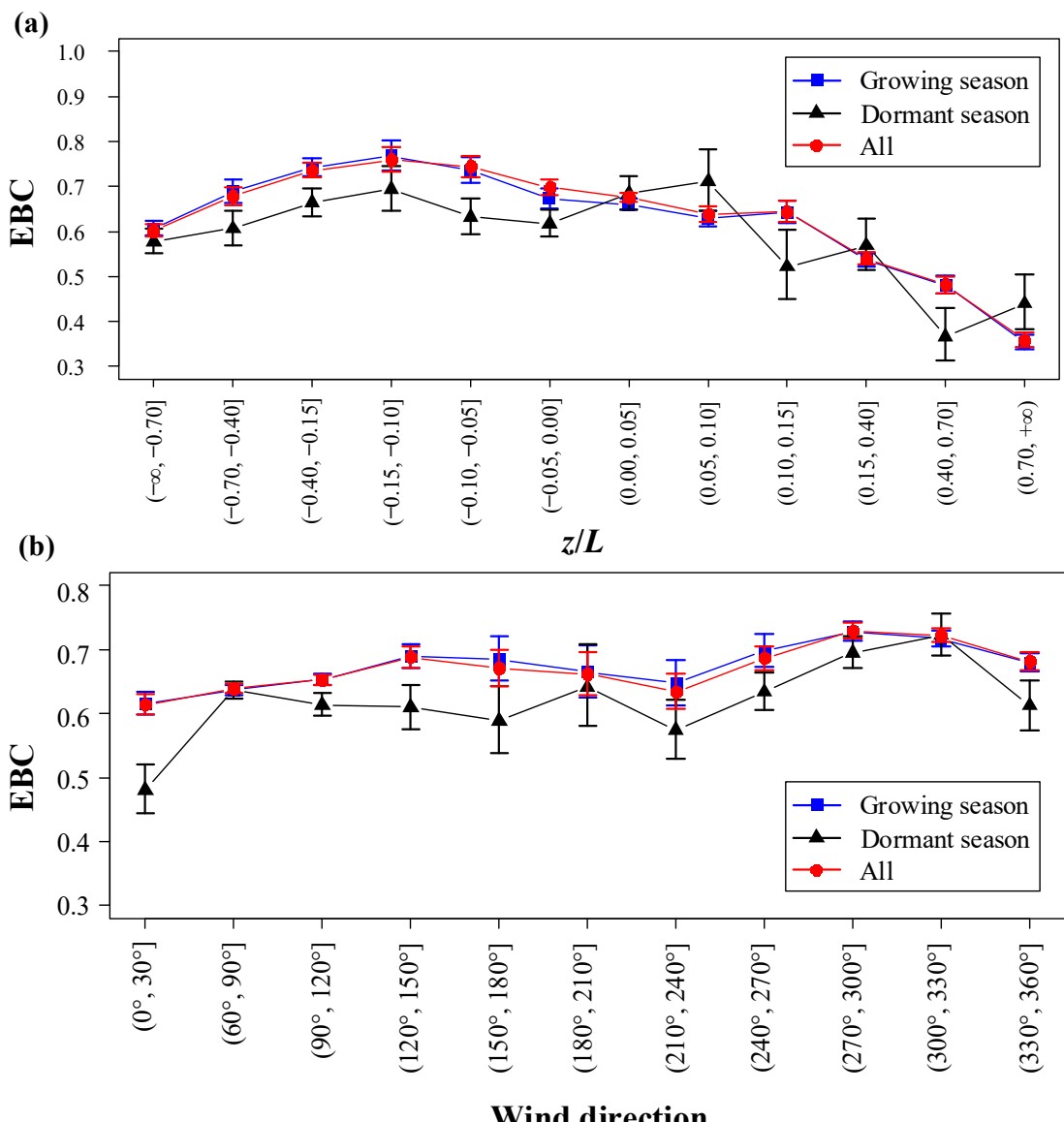

**Figure 12.** Average EBC for each bin of (**a**) atmospheric stability ($z/L$) and (**b**) wind direction.

Average values of EBC are plotted for different wind directions (Figure 12b) to show how the surrounding terrain affects the EC measurements. During both the growing season and dormant season, EBC values were lowest in the bin (0°, 30°]. EBC values during the growing season were highest in the bin (270°, 300°], whereas during the dormant season EBC values were highest in the bin (300°, 330°]. During the dormant season, EBC varied substantially more with wind direction than during the growing season. This suggests that the source area of the flux measurement may be an important cause of interseasonal differences in EBC at this site.

Next, the relationship between TKE and EBC was examined for the two seasons. EBC values clearly increased with increasing TKE during the both the growing season and the dormant season (Figure 13a). However, EBC values were substantially higher during

the growing season than during the dormant season, except when TKE is lower than $0.05 \ \text{m}^2 \ \text{s}^{-2}$ or greater than $1.50 \ \text{m}^2 \ \text{s}^{-2}$. This indicates that EC measurements produce better energy balance closure during the growing season. Nevertheless, while seasonal variations modulate EBC, they do not affect the overall positive relationship between TKE and EBC.

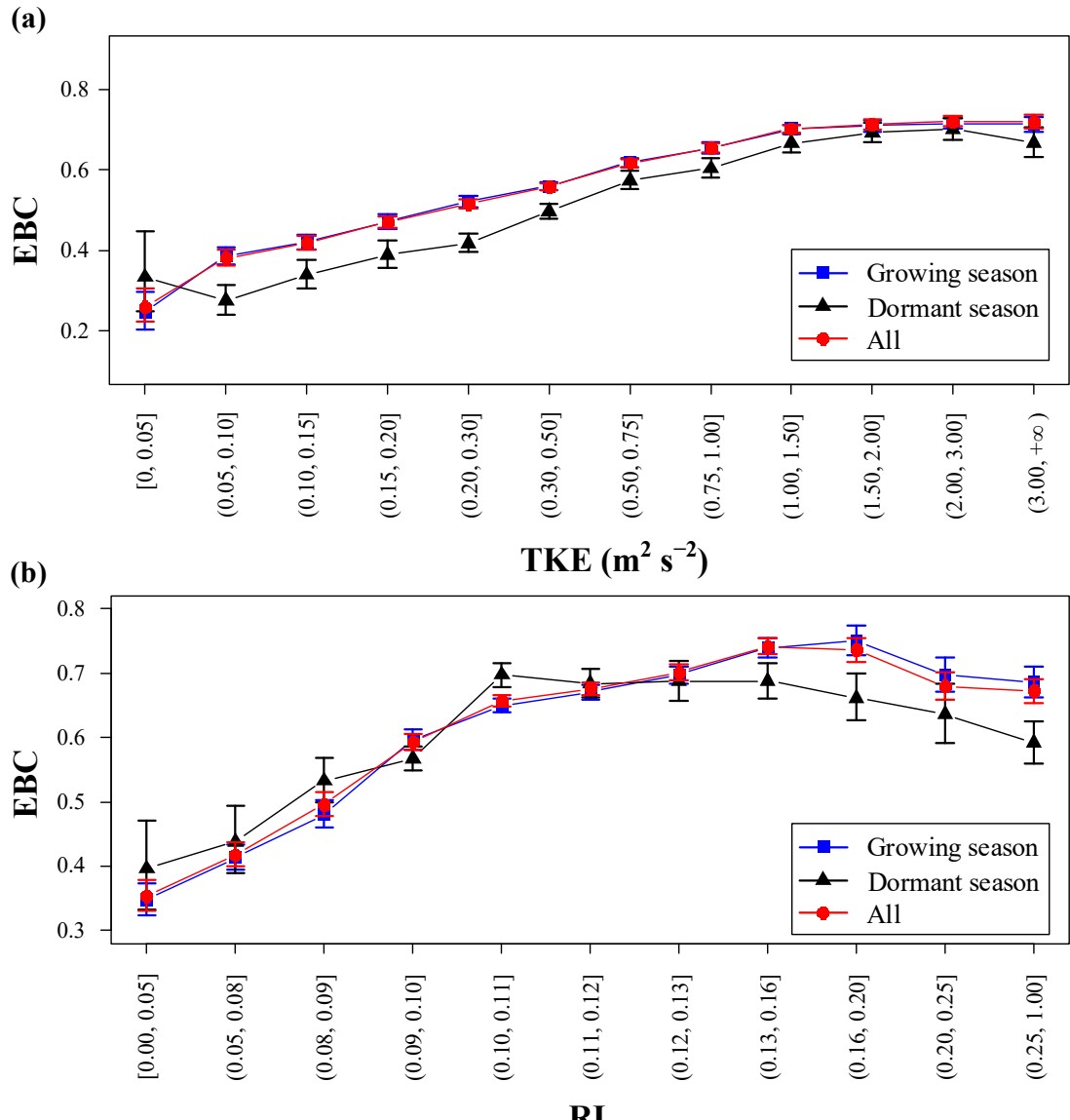

**Figure 13.** Average EBC for each bin of (**a**) turbulent kinetic energy (TKE), and (**b**) relative vertical turbulence intensity (RI).

The role of vertical turbulence intensity in energy balance closure was observed through the relationship between RI and EBC (Figure 13b). When RI < 0.16, EBC increased with increasing RI during the growing season. In the dormant season, a comparable pattern was observed, though EBC peaked at RI values of 0.10–0.11 and subsequently decreased with increasing RI. Lower TKE and RI values are associated with weaker turbulent mixing and therefore smaller EBC values.

During both the seasons, EBC tended to increase with the increasing VPD (Figure 14a); however, during the growing season the increase became more gradual at VPD > 10 hPa. With respect to wind speed, EBC during both seasons increased with increasing wind speed up to 3.0–3.5 m s$^{-1}$, while average EBC changed little with wind speed at wind speed > 3.5 m s$^{-1}$. At most wind speeds, EBC was significantly higher during the growing season than during the dormant season.

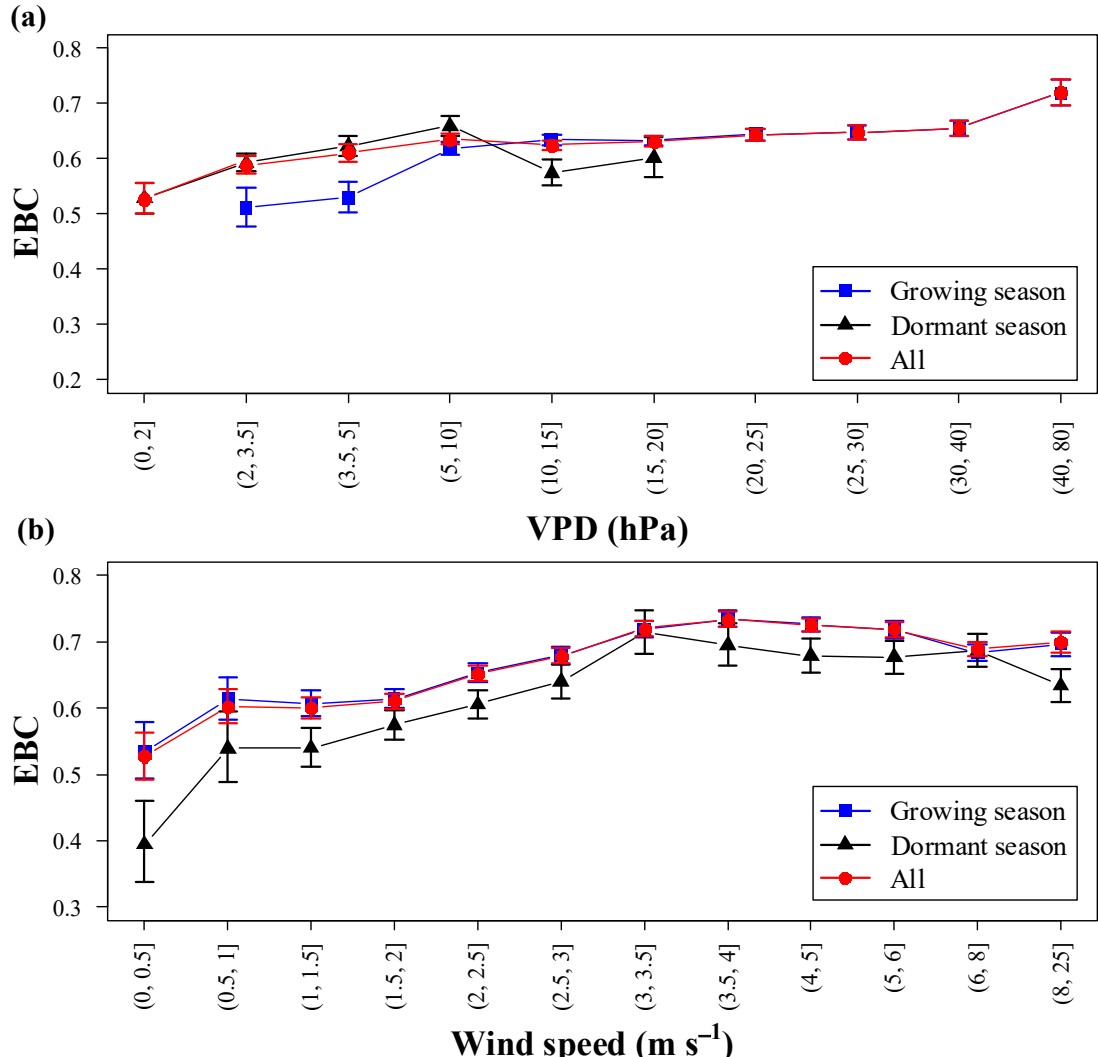

**Figure 14.** Average EBC for each bin of (**a**) vapor pressure deficit (VPD), and (**b**) wind speed. For certain VPD bins, insufficient data were available for either the growing or dormant season, so EBC values for that season were omitted.

To assess the impact of meteorology variation on energy closure, EBC is plotted by air temperature difference ($\Delta T_{air}$) and relative humidity difference ($\Delta RH$; Figure 15). During the growing season, EBC decreased with increasing $\Delta T_{air}$ across a large range of $\Delta T_{air}$ values. A similar trend was observed during the dormant season, but only for $\Delta T_{air} > -0.3$ °C. This suggests that increases in $T_{air}$ during a 30-min period are associated with lower EBC, while decreases in $T_{air}$ during a 30-min period are associated with higher EBC. During both the growing and dormant seasons, EBC increased with increasing $\Delta RH$ up to $\Delta RH$ values of about 0. Thus, greater decreases in RH during a 30-min period are associated with progressively lower EBCs. At $\Delta RH$ values exceeding 2%, EBC drops substantially, indicating that the largest increases in RH during a 30-min period are also associated with poorer energy balance closure.

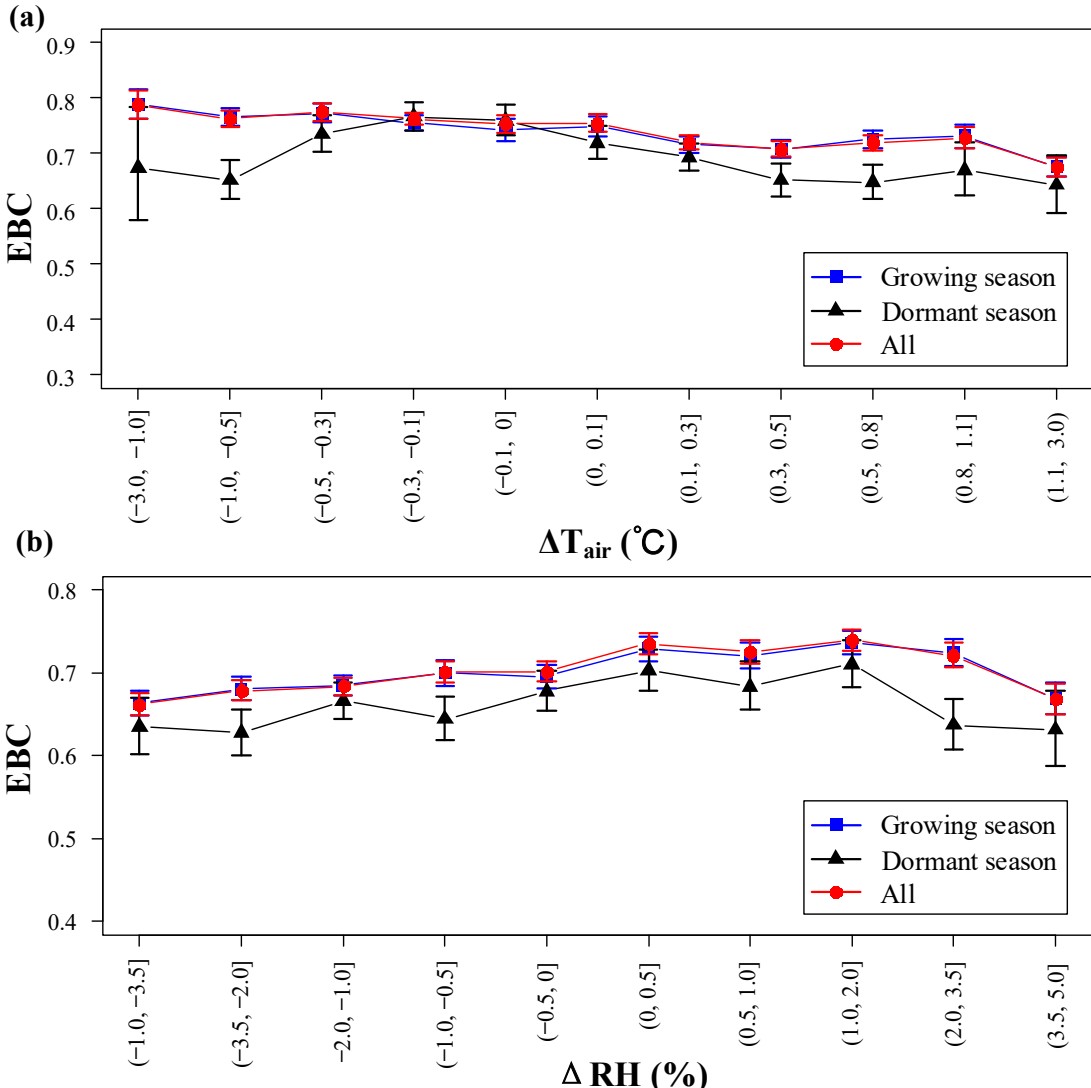

**Figure 15.** Average EBC for each bin of (**a**) 30-min air temperature difference ($\Delta T_{air}$), and (**b**) 30-min relative humidity difference ($\Delta RH$).

## 4. Discussion

### 4.1. Variation in Energy Fluxes

The kinetic energy of the atmosphere is ultimately derived from solar energy, and the atmosphere seldom absorbs solar radiation. Therefore, the surface provides the energy source for the atmosphere by absorbing and converting solar energy. Surface–atmosphere energy circulation produces clear diurnal and seasonal variations in the boundary layer. In a riparian forest, the surface–atmosphere energy circulation is affected by both the nearby river and weather conditions [41]. As a typical desert riparian forest, the Tugai forest experiences the oasis-desert effect [32]. When a strong dry hot wind blows from the desert to the oasis, a surface layer inversion may form over the oasis. This causes sensible heat to be transported downward to the surface (negative sensible heat flux; [32]).

During the growing season, the energy of the desert Tugai forest is mainly used for ecosystem evapotranspiration, and the turbulent flux is dominated by the latent heat flux (Figure 6a). This is consistent with previous studies of many arid and semi-arid areas [42]. The primary reason for this is that the Tugai forest is located near a river and has high vegetation coverage (Figure 1). Thus, evaporation and transpiration are relatively strong, causing greater latent heat transport. The vegetation coverage conditions of the underlying surface play an important role in modulating the surface energy fluxes. During

the dormant season, when vegetation is much sparser and vegetation transpiration is much lower, the turbulent flux is dominated by sensible heat flux, because the latent heat flux is reduced [43], the temperature is lower (Figure 2c), and the energy is used to heat the air.

Within ecosystems, the redistribution of energy from incoming radiation may be affected by weather, temperature, soil, and underlying vegetation surfaces [44]. The average annual potential evapotranspiration (~1500 mm) in the study area is much higher than the average precipitation (~100 mm; [45]). It should be noted that the average annual latent heat flux is less than half of the average net radiation, primarily due to the extremely low annual precipitation.

*4.2. Variation in Energy Balance Closure*

Monin–Obukhov similarity theory provides the theoretical basis for EC flux measurements. However, mesoscale motions can cause the Monin–Obukhov similarity theory to deteriorate or even fail at the stable boundary layer [46]. Therefore, when the measured quantities are affected by mesoscale motions, the EC method will not be able to accurately measure fluxes from boundary layer turbulence [47]. In cases of weak turbulence, some observations will be affected by mesoscale motions [48]. Conversely, for stronger atmospheric turbulent mixing, the EC method assumptions are likely to be closer to reality, resulting in higher EBC values [49].

Based on the above conclusions, the relationship between the relevant variables (i.e., $u_*$, TT, TKE, and RI) and the EBC value can be explained. The data distribution with respect to TT bins showed obviously diurnal and seasonal differences. Some of the nighttime flux data were measured under unstable and high turbulence conditions (Figure 5). This may be one of reasons why the EBC value during the night was higher than that under stable and low turbulence conditions (Figure 8). The turbulence intensity during the dormant season was lower than that during the growing season (Figure 5). The seasonal difference in turbulence intensity under stable conditions was greater than that under unstable conditions, resulting in large differences in EBC values. Although at high values of RI, EBC declines with increasing RI, EBC typically increased with increasing RI (Figure 12). The reason for this may be that when turbulence is weak in the near-surface layer, the EC system cannot fully observe the large-scale latent and sensible heat fluxes [50]. Friction velocity, $u_*$, is an important parameter that represents horizontal atmospheric motion in the near-surface layer [51,52]. When the friction velocity is small, some heat loss during the process of the small-scale movements must occur, resulting poor energy balance closure. This analysis is consistent with previous studies [7,24,26,27]. The positive linear correlation of TKE and EBC in this study shows reasonable agreement with prior simulated and observed results [26,39]. The TKE can be used to correct the energy balance closure [26], but the underlying vegetation and non-turbulent atmospheric motion must also be considered.

Atmospheric stability is an important parameter that represents overall turbulence characteristics. The heterogeneity of the underlying surface is one factor that affects atmospheric stability, with heterogenous surfaces generating shear flows that are rarely homogeneous or stable. The terrain at the measurement site in this study is clearly heterogenous (Figure 6). In heterogenous terrain, some studies hypothesize that large eddies often occur in strongly unstable conditions, resulting in relatively poor energy balance closure [9,11]. Furthermore, these studies observed the highest EBC values under near-neutral atmospheric conditions ($-0.10 \leq z/L < 0$). In contrast, this study found the greatest EBC values in moderately unstable conditions ($-0.15 \leq z/L < -0.10$), which accords with the results of McGloin [24]. $R_{wT}$ and $R_{wq}$ represent the effects of the low-frequency process on the phase difference between the vertical velocity and atmospheric scalars (e.g., T and q). In theory, the values of $R_{wT}$ and $R_{wq}$ can be used to evaluate the EBC value, a result found by Kaimal and Finnigan [53]. However, the effects of low-frequency processes on $R_{wT}$ and $R_{wq}$ vary between measurement sites [26]. This study found a positive linear relationship between absolute $R_{wT}$ and EBC during the dormant season, indicating that

low-frequency processes (i.e., large eddies) tended to decrease $R_{wT}$, which is consistent with von Randow [28]. However, Gao [19] found that low-frequency processes primarily decrease $R_{wq}$, rather than $R_{wT}$. The most energetic large eddies are the relatively important ones for transporting heat.

There is a significant difference in the slope of EBC between the dormant and growing seasons, and that of the growing season is greater than that of the dormant season according to the statistical results of SMA (Figure 9). Under unstable atmosphere and strong turbulence conditions, the seasonal variation in EBC is very small and negligible. EBC is affected by multiple meteorological variables, and some of these are correlated. There are also seasonal differences in the correlation between meteorological variables. The synergistic or antagonistic effects of multiple meteorological variables on EBC may reduce the EBC value.

### 4.3. Causes of Energy Imbalance in This Study

Energy transport in the near-surface layer must satisfy energy and mass conservation. Theoretically, the energy balance of the near-surface layer should be closed; however, the theory only approximates the near-surface energy balance, and it is difficult to achieve balance in observations. The degree of energy balance closure observed by the EC method is determined by the state of atmospheric turbulence. There is usually a close relationship between turbulence and the atmospheric motion on larger scales, and the separation of atmospheric motions into various scales may introduce measurement biases [54]. The energy transfer at the boundary layer, even at the near-surface layer, is not entirely due to turbulence [55]. In this study, the EBC (30-min EBC averages ranged from 64.3% to 67.0%) over the Tugai forest in ELNWNR was moderate compared with observations at similar locations [7,11]. The EBC also varied between seasons and with the wind direction. In the results presented, the energy imbalance problem cannot be ignored at this site. Within the footprint area, the heterogenous terrain influences atmospheric motion and the river generates substantial thermal heterogeneities (Figure 1). These factors may impact near-surface turbulent energy exchange. Thus, research that uses turbulence similarity theory faces serious challenges under untable and heterogeneous conditions, especially when near-surface energy imbalance is observed [56]. Overall, our understanding of energy transfer is relatively good for unstable or strongly turbulent flows; however, our understanding of the near-surface energy transfer mechanism in stable conditions or weakly turbulent flows remains insufficient [46].

Even under stable and homogeneous conditions, energy in the near-surface layer cannot be completely transported by turbulence, causing the observed near-surface energy balance to remain unclosed [57]. However, real surfaces are usually heterogeneous, and the actual turbulence is not completely homogeneous [39,58–60]. Intermittent and coherent structures (including low-frequency eddy motions) will inevitably degrade the accuracy of turbulent flow measured by the EC method, resulting in energy imbalance [61]. For example, unstable atmospheric motion (mainly at sunrise and sunset), horizontal advection caused by heterogeneous terrain, vertical convection, and intermittent turbulence can each substantially affect energy balance [62,63]. Additionally, the horizontal heterogeneity of latent heat flux affects the energy balance, and the mesoscale circulation induced by surface heat flux variations will cause additional flux transport [64]. Greater surface heat flux variations will cause larger heat fluxes from mesoscale motions. In this study, the secondary circulation between the forest and the desert contributed to the energy imbalance at the EC site [65]. Moreover, the spatial and temporal non-stationarity of latent heat flux causes significant variation in energy balance closure [5,41]. Key challenges include quantitatively characterizing the strength of these factors and evaluating how their effects on near-surface energy balance vary with the strength of turbulence. Finally, under certain conditions, free convection can also occur in the near-surface layer [66,67].

Another important factor that contributes to near-surface energy balance is instrument measurement error. The EC flux footprint area differs from the measurement areas of

the Rn, G, and S instruments. In theory, the underlying surface areas measured by the net radiation meter and the EC instrument are unlikely to be consistent [60,68,69]. If the underlying surfaces measured by the EC instrument and the energy meter have substantial heterogeneity (e.g., an open canopy or a multi-component canopy), the mismatch between measurement areas will increase energy balance closure errors [39], especially when other energy uptake terms (e.g., photosynthetic energy) are ignored. Even if the five energy terms (LE, H, Rn, G, and S) can be accurately measured, the energy balance cannot be completely closed, because other energy uptakes exist in the energy balance system [5]. These uptakes include meteorological processes, such as melting, freezing, and sublimation. They also include the thermal storage of soil heat flux in the upper soil layer, thermal storage in canopy vegetation, and the photosynthetic energy conversion of plants [7,9]. In this study, these energy uptake terms are not considered, except for canopy heat storage; ignoring these terms likely adds some error to the estimation of effective energy.

We have inserted the slope term of SMA regression in Figure 8 to represent the diurnal variations in EBC. Since our model does not force the intercept term to be zero, the intercept term of the SMA regression exhibits a diurnal variation——less than zero during the daytime and greater than zero during the nighttime. The intercept term is less (or greater) than zero, indicating that the LE + H observation is less (or greater) than the true value, or that the R − G − S observation is greater (or less) than the true value. The scattered points are mainly distributed in the first quadrant during the daytime and distributed in the third quadrant during the nighttime. Considering the atmosphere conditions, the observations of LE + H and Rn during the daytime are relatively reliable. It can be inferred that the absolute value of G + S observations is less than the true value. Due to research funding restrictions, part of the flux storage term S was not measured and calculated in this study. Obviously, this part is affected by RH and temperature changes, resulting in EBC variation. This is consistent with the results in Figure 15. Beyond affecting near-surface heat storage and latent heat transfer [70], the complexity of the soil environment affects the calculation of G [71]. Previous studies showed that the magnitudes of G measured by the heat flux plate is relatively small [72]. The soil heat flux calculation method in this study used shallow soil (0−5 cm) observational data to calculate surface soil heat flux. This layer is quite thin, so its temperature and moisture vary substantially in space and time. The effects of variations in seasonal soil conditions and vegetation cover produce great uncertainty in the measurement and calculation of shallow soil heat flux [73].

The near-surface energy imbalance problem will be completely solved only through a large number of in-depth and meticulous observational studies. These studies must also pay attention to the universality and particularity of near-surface turbulent energy transport.

## 5. Conclusions

This study used 7 years of eddy covariance data from a flux observation station in ELNWNR in Xinjiang, Northwest China, to analyze variations in turbulent flux, available energy flux, and energy flux closure in the Tugai forest. Radiation flux, sensible heat flux, latent heat flux, and surface soil heat flux in the Tugai forest experience substantial daily and seasonal changes. The radiant flux and turbulent flux regularly showed minimum values at night and a single maximum value at noon. Across the growing and dormant seasons, the diurnal variations of radiant flux and turbulent flux were consistent. However, there was significant seasonal variation in energy flux distribution, with latent heat flux dominant during the growing season and sensible heat flux dominant during the dormant season.

Significant diurnal variation in EBC was observed in the Tugai forest. The EBC estimated by SMA ranged from 0.64 to 0.67, with different phenomena causing surface energy imbalance during the growing and dormant seasons. The state of atmospheric turbulence, which includes $u_*$, RI, TKE, and atmospheric stability, is the most important factor affecting EBC. Additionally, meteorological factors, such as wind direction, the

amount of non-turbulent processes, and the oasis-desert effect, modulated the measured EC values. Furthermore, the energy balance was affected by vegetation cover and topographical heterogeneities. This study only used observational data from a single flux site, so its conclusions are preliminary. Given the uncommon topography and surface thermal properties of the flux observation station, the universality of the results presented in this study must be verified by more extensive observations at diverse locations.

**Supplementary Materials:** The following are available online at https://www.mdpi.com/1999-4907/12/2/243/s1, Table S1: average wind speed (m s$^{-1}$) and wind direction; Table S2: flux data included H (W m$^{-2}$), LE (W m$^{-2}$), Rn (W m$^{-2}$), G (W m$^{-2}$), S (W m$^{-2}$) and meteorological data.

**Author Contributions:** D.T. analyzed the data and wrote the paper; G.L. conceived the study and revised the paper; X.H. and L.Q. collected the data. All authors have read and agreed to the published version of the manuscript.

**Funding:** This work is supported by the Natural Science Foundation of Xinjiang Uygur Autonomous Region (Grant No. XJEDU20201002) and the National Natural Science Foundation of China (Grant No. 31560131 and 31760186).

**Institutional Review Board Statement:** Not applicable.

**Informed Consent Statement:** Not applicable.

**Data Availability Statement:** Data supporting reported results are available in supplementary materials.

**Acknowledgments:** We thank Jinlong Wang and Jingzhe Wang for the initial illustration of Figure 1. We are grateful to anonymous reviewers and handling editors for their insightful comments which greatly improved an earlier version of this manuscript.

**Conflicts of Interest:** The authors declare that they have no conflict of interest.

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
