# Peer review of "Energy Balance Closure in the Tugai Forest in Ebinur Lake Basin, Northwest China"

_forests, doi:10.3390/f12020243_

Round 1

Reviewer 1 Report

This paper reports energy balance closure (EBC) in tugai forest for seven years and examined environmental factor to influence EBC. Study desert forest is less studied and hence analysis of long-term energy flux data is meaningful to report. However, authors provide many analysis results without explaining. And more importantly, EBC should be calculated as mean (H+LE)/mean(Rn-G-S) rather than mean [(H+LE)/(Rn-G-S)] to exclude the effect of outlier (Zhou et al., 2019). Therefore, current results are not consistent with previous results (Zhou et al., 2019) and it is difficult to explain physical meaning for the behavior of bin averaged EBC in terms of other variables. I suggest to recalculate EBC and analyze them.   

Specific comments

  • Line 85: conducted -> calculated
  • Provide mean canopy height in 2.1 study site description,
  • Line 103: use W m-2 for radiation unit
  • Line 169, formulation is  wrongly written, dimension does not match.
  • Line 191, how did you calculate displacement height?
  • For all variables in the text, use italic
  • In Fig.2, for 2017-2019, precipitation observation is missing? If so, I suggest to remove the precipitation figure because half of data are missing and hence the figure does not give information on annual variation of precipitation.
  • In Fig 6, in some wind direction, reported z0 reaches 12m, it is too high (unreasonable) over forest? What is alphabet in figure 6? There is no explanation on this in caption of Fig. 6.
  • Line 245-246, a significant 8difference? What does that mean? 8 wind directions show significant difference in z0? How did you evaluate significance?
  • Lines 262, 263 at many lines, Instead of AM, PM, use LST
  • Both Fig. 7 and Fig. 8 are unnecessary, combine Fig 7 and Fig 7 by adding EBC in Figure 7.
  • In Fig. 10, Why does EBC show low EBR in Rwt near 0 during dormant season but not growing season? Explain it.
  • In Figure 15, during dormant season, dT and EBC does not correlate and dRH and EBC does not show correlation
  • In Fig. 8, during nighttime, Rn-G-S has larger magnitude than H+LE, but the EBC ratio is close to 1 at night during growing season? How is it possible? Have you calculated EBR using mean energy fluxes Rn, G, S, H, and LE. Such calculation is more reasonable
  • In Fig. 9, Have you try plotting separately for daytime and nighttime? Nighttime figure shows better correlation between Rn-G-S and H+LE? Fig. 8 shows better EBC during nighttime.
  • Line 348, EBC tends to increase with the increasing VPD. Do you think that this is not coincidence but physically meaningful? explain why EBC increases with increasing vpd?
  • In Lines 377-380, author mention oasis effect in a typical desert forest but this result does not show such oasis effect. Why do you mention this here? Oasis effect occurred at this site?
  • Line 430, Kaimal and Finnigan (1994) is missing in reference.

Reviewer 2 Report

The reviewed work “Energy balance closure in the Tugai forest in Ebinur Lake Basin, Northwest China” relates to among others characteristics of the energy flux, energy balance closure (EBC), and the factors that influence EBC. The topic is suitable for Forests. The results are new and interesting, and they can be considered as basis for future studies that should be conducted in other areas of China. The text is well written, but could be improved by some corrections and accurations. Therefore, my recommendation for the manuscript is a minor revision. Detailed comments are listed below:
Title:
- The title is ok
Abstract:
- The authors could briefly mention here can briefly mention the practical possibilities offered by the methods used by the Authors.

Introduction:

- Have other authors (please quote, give examples) considered the problem of closing the energy balance in relation to finding physical factors and processes and in relation to quantitative analyzes of these factors and processes? (lines 71-74)

Materials and methods

- I have no objections. The chapter discussing materials and methods was written correctly, and all the details were included. The project was very extensive and included many variables.

Results

- I have no objections. Multifaceted figures testify to the reliability of the obtained results. Despite the very nature of the work, these results shed new light on the study closing the energy balance in a forest in China.

Discussion:

- The discussion is extensive about the taken subject and the results obtained. However, the authors do not cite too many research groups showing similarities between the studies. Please complete this.

Author Response

Dear Reviewer:

Thank you for your letter and for the reviewers’ comments concerning our manuscript “Energy balance closure in the Tugai forest in Ebinur Lake Basin, Northwest China”. We have studied comments carefully and have made correction. Those comments are all valuable and very helpful for revising and improving our paper, as well as the important guiding significance to our researches. The purpose of this letter is to address comments. We have studied comments carefully and have made correction point by point which we hope meet with approval. Revised portion are marked in red in the manuscript. The corrections in the manuscript and the responses to the reviewer’s comments are as following, in the responses, the numbers of references and lines refer to those in the revised manuscript.

Reviewer comment No. 1: Have other authors (please quote, give examples) considered the problem of closing the energy balance in relation to finding physical factors and processes and in relation to quantitative analyzes of these factors and processes? (lines 71-74)

Response to Reviewer comment No. 1: Thanks for the comment. Many researchers have focused on EBC problem and hence much knowledge on this problem related to physical factors and processes. We added the sentence "Some preceding studies demonstrated that EBC was a function of environmental factors, which has been confirmed by many observational studies (Cui and Chui, 2019; McGloin et al., 2018; Zhou and Li, 2019). " to Introduction.

Reviewer comment No. 2: The discussion is extensive about the taken subject and the results obtained. However, the authors do not cite too many research groups showing similarities between the studies. Please complete this.

Response to Reviewer comment No. 2: Thanks for the comment. In order to discuss the results in depth, we have searched a large number of related papers. For the content of the discussion, many recent works were cited. We have revised this paper several times in accordance with the comments and suggestions of multiple reviewers (i.e. AFM, Ecosystems), and believe that the cited studies are sufficient.

References

Cui, W. and Chui, T.F.M., 2019. Temporal and spatial variations of energy balance closure across FLUXNET research sites. Agricultural and Forest Meteorology, 271: 12-21.

McGloin, R. et al., 2018. Energy balance closure at a variety of ecosystems in Central Europe with contrasting topographies. Agricultural and Forest Meteorology, 248: 418-431.

Zhou, Y. and Li, X., 2019. Energy balance closures in diverse ecosystems of an endorheic river basin. Agricultural and Forest Meteorology, 274: 118-131.

Round 2

Reviewer 1 Report

Many comments were revised in the manuscript. But major concern is inconsistency of the results.

Major comments

Figure 8 shows high EBR during night time. But Figure 11, 12a, 13, 14 indicate low EBR in stable and low turbulence condition which is character of nighttime (Fig. 5). Previous studies show low EBR during night time and low EBR is related to low turbulence and stable condition. But your study shows opposite (high EBR during nighttime) but EBR show similar relationship with turbulence and stability parameter to previous studies. How can you explain this? In the discussion, you mentioned that low EBR is related to low turbulence in stable condition, but this is contrary to Fig. 8. You need to provide explanation for that.  

Lines 438-441: Is this the reason for EBC decrease with increasing RI at high RI? High RI indicates turbulent condition but you explain weak turbulence condition. I think that sentence should be “Although at high value of RI, EBC declines with increasing RI, EBC typically increases with increasing RI. The reason for this ..” 

Figure 9 shows similar slope of EBC for dormant and growing season. But Fig.10b and 11b shows large difference of EBC between growing and dormant seasons, particularly in Fig 11b. Explain it.

Figure 11b shows that in unstable condition (TT>0) EBC in dormant season is lower than that during growing season by about 0.3 but Figure 12a, in unstable condition (z/L<0) EBC during dormant season is lower than that during growing season by 0.1. In Fig 11b, EBC in unstable condition (TT>0) is lower than 0.5 in most bins during dormant season but EBC is higher than 0.5 in most bins in unstable condition during dormant season. Why is EBC value so different?

Minor comments

Lines 456, 459, “unsteady” should be “unstable”.

Variable should be written in italic (ex, lines 178, 179, 186, 187 etc in many lines), .

For von Karman constant, different notation is used (k (line 178) and  (line 199) ).

Line 199, von Karman

Line (197), u should be . 

In Figure 4: count number should be on the circle
